# Simplifying drone-based aboveground carbon density measurements to support community forestry

Ben Newport[1]*, Tristram C. Hales[2], Joanna House[1,3], Benoit Goossens[4,5,6], Amaziasizamoria Jumail[4,5]

1 School of Geographical Sciences, University of Bristol, Bristol, United Kingdom, 2 School of Earth and Environmental Sciences, Cardiff University, Cardiff, United Kingdom, 3 Cabot Institute, University of Bristol, Bristol, United Kingdom, 4 Organisms and Environment Division, School of Biosciences, Cardiff University, Cardiff, United Kingdom, 5 Danau Girang Field Centre, c/o Sabah Wildlife Department, Kota Kinabalu, Malaysia, 6 Sabah Wildlife Department, Kota Kinabalu, Malaysia

* ben.newport@bristol.ac.uk

## Abstract

Community-based forest restoration has the potential to sequester large amounts of atmospheric carbon, avoid forest degradation, and support sustainable development. However, if partnered with international funders, such projects often require robust and transparent aboveground carbon measurements to secure payments, and current monitoring approaches are not necessarily appropriate due to costs, scale, and complexity. The use of consumer-grade drones in combination with open source structure-from-motion photogrammetry may provide a solution. In this study, we tested the suitability of a simplified drone-based method for measuring aboveground carbon density in heavily degraded tropical forests at a 2 ha restoration site in Sabah, Malaysia, comparing our results against established field-based methods. We used structure-from-motion photogrammetry to generate canopy height models from drone imagery, and applied multiple pre-published plot-aggregate allometric equations to examine the importance of utilising regionally calibrated allometric equations. Our results suggest that this simplified method can produce aboveground carbon density measurements of a similar magnitude to field-based methods, quickly and only with a single input metric. However, there are greater levels of uncertainty in carbon density measurements due to errors associated with canopy height measurements from drones. Our findings also highlight the importance of selecting regionally calibrated allometric equations for this approach. At scales between 1 and 100 ha, drone-based methods provide an appealing option for data acquisition and carbon measurement, balancing trade-offs between accuracy, simplicity, and cost effectiveness and coinciding well with the needs of community-scale aboveground carbon measurement. Of importance, we also discuss considerations relating to the accessibility of this method for community use, beyond purchasing a drone, that must not be overlooked. Nevertheless, the method presented here lays the foundations for a simple workflow for measuring aboveground carbon density at a community scale that can be refined in future studies.

**Data availability statement:** All drone images and field data are publicly available from the CEDA Archive, a NERC repository for earth observation data. The dataset can be accessed via the following catalogue record link: https://catalogue.ceda.ac.uk/uuid/98692e-c457ee431cacc4027820e46411/ (DOI: https://doi.org/10.5285/98692ec457ee431c-acc4027820e46411).

**Funding:** BN was funded by a NERC GW4+ Doctoral Training Partnership studentship (NE/L002434/1) from the Natural Environment Research Council (www.ukri.org/councils/nerc/) and an SWDTP studentship (ES/P000630/1) from the Economic and Social Research Council (www.ukri.org/councils/esrc/). The funders played no role in the study design, data collection and analysis, decision to publish, or preparation of the manuscript.

## Introduction

Small-scale, community-based forest restoration can sequester large amounts of atmospheric carbon, reduce emissions from deforestation and degradation, and support sustainable development [1–5]. Community-scale projects typically cover tens of hectares or less and are implemented by stakeholders including community groups, villages, and NGOs. Numbering in the tens of thousands globally, such projects are important for two key reasons. Firstly, they involve indigenous and rural communities in forest management, which is a key factor in enhancing both the ecological and social outcomes of restoration activities [6–8]. Empowering communities increases local engagement with projects [9], incorporates local knowledge, and assures rural populations receive their desired benefits from global restoration initiatives [5]. Secondly, forests restored in this manner are more likely to persist into the long-term (decades to centuries) than large-scale tree planting projects developed without community support [10,11]. Industrial carbon sequestration projects can fail due to poor site and species selection, mismanagement, and an over-focus on planting versus long-term maintenance [12–15], leading to negligible changes in canopy cover or carbon storage [16,17]. By accommodating local knowledge and needs, such as the provision of food or firewood, community-scale projects are able to avoid these pitfalls, increasing forest cover and maintaining long-term local support [18].

Many community-scale projects partner with funders from developed nations who provide financial compensation to support climate and sustainability-oriented goals such as carbon offsetting. These financial mechanisms require projects to provide robust biomass measurements to verify baseline carbon values at restoration sites [19,20]. However, current established methods for measuring aboveground carbon density (ACD, often reported in Mg C ha$^{-1}$) are not necessarily appropriate for use at the community scale, are time consuming, and require specialist training.

Several methods are currently used to quantify ACD in forest stands including field-based or remote sensing surveys of tree metrics. Remotely-sensed variables are used to calculate ACD via a series of empirical allometric equations, which predict tree biomass from easier-to-measure variables such as height or diameter and are supported by statistical analysis based on ACD values from permanent field plots [21–24]. The increasing availability and accessibility of remote sensing data make this an important tool for forest restoration. The benefit of using remote sensing is that it can be employed over large scales and in remote areas, and is often cheaper and more feasible than extensive ground surveys. While such an approach has been employed extensively by academics and commercial foresters, it presents challenges for use at a community scale. The cost of procuring high-resolution (<3 m) remote sensing imagery suitable for community-scale carbon quantification can be prohibitively expensive for community-scale actors. Freely available datasets (e.g., Landsat, GEDI) may have too coarse a resolution for meaningful or timely analysis, with low resample rates exacerbated by persistent cloud cover in the tropics [25,26].

Lightweight, low-cost, consumer-grade drones (also known as unmanned [sic] aerial vehicles (UAVs) [although see [27]]) offer a potential solution to these data acquisition issues. Consumer-grade drones are relatively cheap (to purchase and to operate) compared to other data collection methods; they can be piloted with minimal training and a smartphone; they have high spatial and temporal resolution; and they grant autonomy over data collection, an important step in empowering and engaging local people in conservation initiatives [28,29]. In addition, the optical imagery that drones generate can be combined with structure-from-motion (SfM) photogrammetry – which produces 3D point clouds from sets of overlapping 2D images [30] – to calculate canopy height and, subsequently, carbon values in a similar manner to other remote sensing approaches [31–33]. Drone-based SfM is a good potential fit for community-scale ACD measurement as it does not require information on camera

location and orientation, enabling the use of inexpensive platforms and sensors [30,34,35]. However, remote sensing-based ACD quantification methods often involve generating novel allometric equations [24,36] which may be challenging for community-scale projects with low levels of external support. The use of pre-published allometric equations offers an alternative option and they are frequently used in field-based individual tree crown (ITC) measurements, either out of convenience or necessity [37]. Yet, to date, there have been few studies investigating the accuracy and uncertainties surrounding the use of pre-published plot-aggregate allometric equations with drone-derived SfM data for small-scale ACD measurements.

In this study we assess the suitability of a simplified method for measuring ACD within the context of community-scale forest restoration, using a consumer-grade drone and open source SfM software. We compare our results against field-based measurements of ACD to examine their biases and uncertainties. We use a restoration site in Sabah (Malaysian Borneo) as a case study site, representing a real-world restoration project where this method would be applicable. In this context, this study not only fills a gap in the literature regarding drone-based ACD measurements at the community scale, but also contributes to practical insights for restoration practitioners in tropical forest restoration.

## Materials and methods

### Study region

We calculated different drone-derived carbon metrics within a 2 ha forest restoration plot in the Pin Supu Forest Reserve (4,696 ha), part of the Lower Kinabatangan Wildlife Sanctuary, Sabah, Malaysia (5°25′15″ N 117°58′05″ E) (Fig 1). The restoration site, known as Kaboi Lake, is managed by the charity Regrow Borneo (www.regrowborneo.org), the Danau Girang Field Centre (DGFC), and the Community Ecotourism Co-operative of the Batuh Puteh Community (KOPEL). Located within the Kinabatangan floodplain, the site is a seasonally flooded freshwater swamp forest. The site has an average annual rainfall of 2700 mm with an average temperature of 25.7 °C [38], and total relief across the site is <1 m. Kaboi Lake lacks any dipterocarps (Dipterocarpaceae family), a numerically dominant and carbon-dense tree family in Borneo [39,40], due to selective logging in the 1980s [41]. Kaboi Lake and the surrounding forest were gazetted by the Sabah Forestry Department (SFD) in 1984 and have since been left to regenerate naturally [42]. In 2020, KOPEL team members cleared the site for replanting, removing elephant grass (*Pennisetum purpureum*), climbing bamboo (*Dinochloa spp.*) and various vines to make way for flood-resistant Bongkol (*Nauclea spp.*) and other native saplings. Fig 1 shows areas of bare soil where clearing took place. Within the restoration site a 50 m x 50 m botanical plot was also established, which we used to compare drone- and field-derived ACD measurements. The project received permission to conduct drone surveys and field data collection in Pin Supu from the Sabah Biodiversity Centre (SaBC) (access license number JKM/MBS.1000–2/2 JLD.11 (11)).

Kaboi Lake is located in the Lower Kinabatangan Wildlife Sanctuary in eastern Sabah, Malaysia, at the northern end of the Southeast Asian island of Borneo (see inset maps). Red line indicates the 2-ha restoration site boundary; green line indicates the 50 m x 50 m botanical plot boundary.

### Aboveground carbon density measurements from drone data

**Drone data collection.** We collected drone imagery of the Kaboi Lake site on 22nd March 2021 using a DJI Phantom 4 Pro V2.0 quadcopter equipped with a 20-megapixel optical camera (DJI, Shenzhen, China). Flight planning was conducted with a tablet and DroneDeploy planning software (www.dronedeploy.com). The flights were fully autonomous and followed two 'lawnmower' patterns, overlapping at 90°, to increase redundancy and reduce occlusions for the SfM processing [35]. Flight altitude was set at 70 m above ground

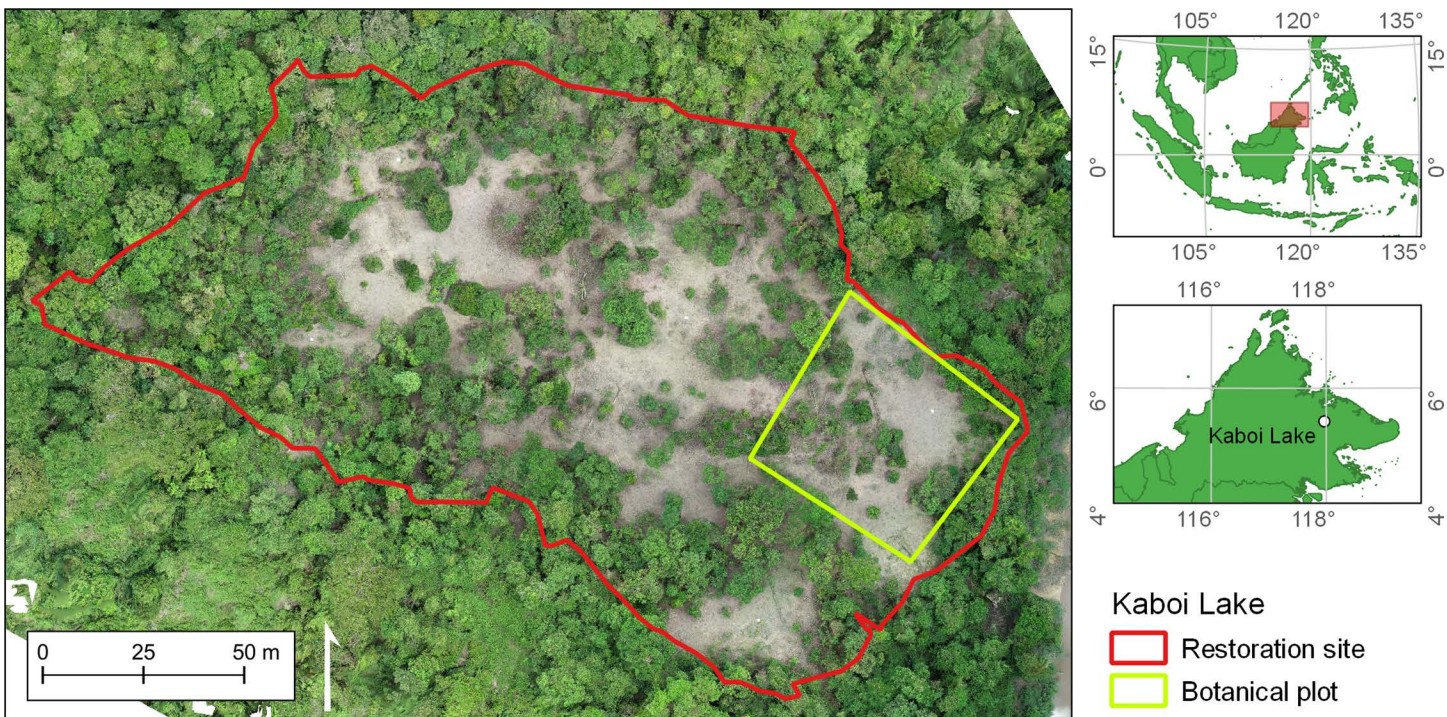

**Fig 1. Orthomosaic of the Kaboi Lake restoration site.** Kaboi Lake is located in the Lower Kinabatangan Wildlife Sanctuary in eastern Sabah, Malaysia, at the northern end of the Southeast Asian island of Borneo (see inset maps). Red line indicates the 2 ha restoration site boundary; green line indicates the 50 m x 50 m botanical plot boundary.

level, resulting in a ground sampling distance of approximately 5 cm, with a flight speed of 5 m s⁻¹ and front and side image overlap of 80%. Three flights of ≈15 minutes each were required to gather a total of 597 images for the 2 ha site.

**Structure-from-motion processing of images.** We performed all SfM image processing using OpenDroneMap (ODM) [43], an open source software ecosystem developed for processing aerial imagery. ODM utilises embedded Exchangeable Image File Format (EXIF) tags within image files to access information on geolocation and camera parameters. The processing pipeline in ODM consisted of five key processes and algorithms [44]: structure-from-motion, producing a spare point cloud; multi-view stereo, generating a dense point cloud; meshing, to create 3D polygonal surfaces from the point cloud; texturing, to then colour the polygons using the relevant input images; and finally georeferencing, which transforms the local coordinate system using geolocation data embedded in the input images.

We conducted all processing on a desktop PC with an Intel Core i7 CPU and 16GB RAM, although more memory is recommended for processing >200 images [44]. All ODM parameters were left as default apart from the following two: input images were resized to a width of 4096 pixels (from 4864) to decrease processing time whilst maintaining high resolution; and the minimum number of features to be extracted from each image for matching in the SfM process was increased from 8,000 to 28,000 due to the lack of distinguishable features in forest canopies. Processing 597 images took 3.5 hours.

**Point cloud processing into canopy height models.** Adapting the workflow outlined by Mlambo et al. [45], we post-processed the georeferenced point cloud using the LAStools suite of LiDAR processing tools [46] in QGIS (version 3.14.16) [47]. Several steps were required to produce a digital elevation model (DEM), digital surface model (DSM), and canopy height

model (CHM) from the data, as outlined in Fig 2. Due to the file size limitations of LAStools algorithms, the point cloud was first split into smaller tiles and then cleaned with the *lasnoise* tool. *Lasnoise* identifies and removes isolated points that have few other points within a three-dimensional search grid centred on that respective point. Cleaned points were then classified as either ground or non-ground returns using *lasground* and *lasclassify*, tools developed for extracting bare-earth points from airborne LiDAR data. The tiles were then thinned, with only the highest points within a 0.05 m x 0.05 m grid (half the intended final resolution) being used to generate DEM tiles, and with only the lowest points used for DSM tiles. Finally, the tiled DEM and DSM rasters were merged to create a single DEM and DSM for the whole site, both at 0.1 m resolution.

The DEM produced in the previous step was very uneven, especially towards the edges of the target site and in places where vegetation cover was high, which did not correspond with the known minimal relief across the site. To resolve this issue, we produced a planar, flat DEM by taking the 15th percentile value of the original DEM as a proxy for the true ground elevation across the site. We verified this assumption by examining the histogram of values for the original DEM and confirming that the chosen ground elevation was a peak value – the most common elevation was very likely to be the floodplain surface given the large areas of exposed ground at the site (Fig 1). This approach has been previously used to generate DEMs in other biomass studies of similar tropical forests with little relief, such as mangrove areas [31]. We created a CHM raster layer by subtracting the flat DEM from the DSM (Fig 2), thereby normalising the heights of the DSM.

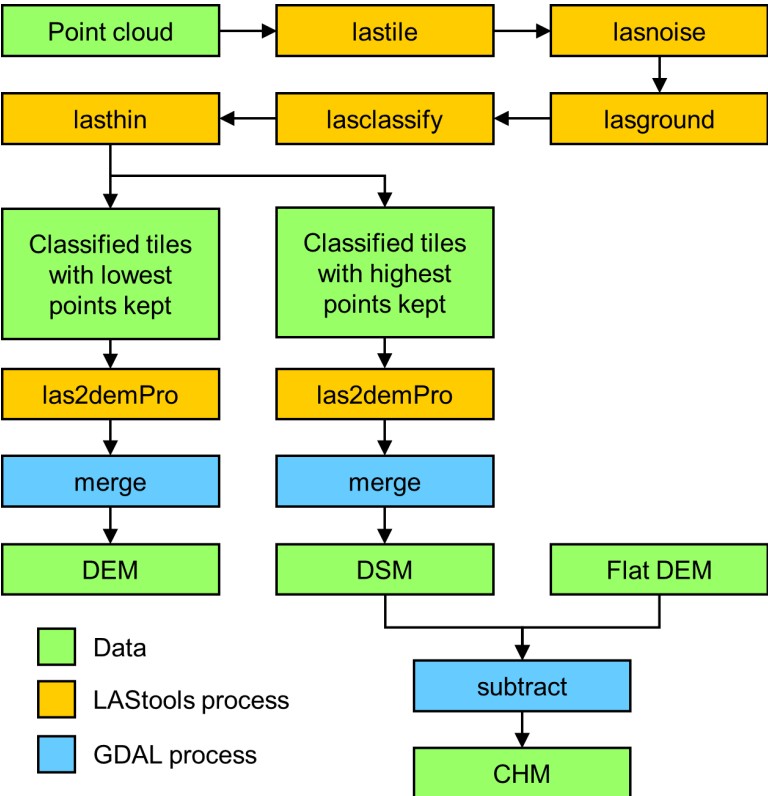

**Fig 2. Workflow for creating a canopy height model (CHM) from point cloud data.**

**Validating the canopy height model.**   We validated the CHM-derived height values by comparing them to field-measured tree heights within the botanical plot (field methods described below). Although the trees in the botanical plot had been surveyed, no geolocation information was recorded, preventing direct extraction of specific tree heights from the CHM. To overcome this, we located individual trees within the CHM using the Python package PyCrown [48]. PyCrown uses local maxima within the CHM to locate tree top positions and delineates tree crowns using region-growing algorithms adapted from [49]. We produced five different estimates of tree numbers and locations using various input parameters, as outlined in S1 Text. We used multiple estimates because field measurements could not be matched directly to the CHM, and different input parameters resulted in over- or underestimates of tree numbers in the botanical plot. Aside from those in S1 Text, all other PyCrown settings were left as default.

**Error propagation in canopy height models.**   Biomass measurements from allometric equations are subject to various sources of uncertainty, from model parameter estimates to field measurement errors. These errors are thought to represent over 20% of the measured biomass at a plot level [50,51]. To account for uncertainties in drone-derived measurements of biomass (and therefore carbon), we first calculated the mean top-of-canopy height (TCH in m), a key value for the plot-aggregate equations used below, by averaging the pixel values within the CHM for the botanical plot. We propagated uncertainty using the Monte Carlo method. Root mean square errors (RMSEs) associated with drone measurements of canopy height can range from less than 0.5 m [52,53] to over 5 m [32,54], though sparse ground coverage [55] and lower canopy heights (<24 m) [54,56] generally contribute to more accurate results. Since the botanical plot had relatively small trees (<20 m) and large areas of bare ground (leading to potentially more accurate measurements), we used two separate error distributions to model different measurement error scenarios: one smaller error distribution with a small standard deviation ($\sigma = 1.5$ m) and a more conservative distribution with larger errors ($\sigma = 4$ m). 1,000 values of mean TCH were generated using each error distribution, yielding 2,000 values for mean TCH for the botanical plot.

**Plot-aggregate allometric equations.**   From a literature review, we identified five suitable plot-aggregate allometric equations to generate ACD measurements from the drone-derived CHM (Table 1). Equations *I* [57] and *IV* [36] are simple power functions which suggest a relationship between canopy height and ACD, and calculate ACD from mean TCH. Equation *I* was calibrated with data from pantropical forests and equation *IV* was based on samples from peat swamp forests in Kalimantan. Equations *II*, *III* [58] and *V* [24] are differently calibrated versions of an additional model developed by Asner and Mascaro [57], in which ACD is measured using TCH as well as estimates of basal area (cross-sectional area of all stems; BA in $m^2$ $ha^{-1}$) and wood density (WD in g $cm^{-3}$). To apply these equations to areas where measurements of basal area and wood density are not available, sub-models are used to calculate BA and WD from TCH, meaning ACD can be measured using the single metric TCH. Equations *II* and *III* were calculated by fitting data from 36 forest plots in Kabili-Sepilok Forest Reserve, a remnant of old-growth tropical forest in eastern Sabah, to Asner and Mascaro's [57] generalised model. Equation *II* used sub-models to estimate BA and WD from TCH, while equation *III* used field measurements instead (equations in Table 1 simplified by authors). Equation *V* was calibrated using plot inventories from five forest reserves across the state of Sabah (including Kabili-Sepilok Forest Reserve), and used sub-models to estimate BA and WD. We applied the five equations to the 2,000 mean TCH values, resulting in 10,000 separate plot-aggregate ACD measurements for the botanical plot, which were categorised by both the degree of error associated with height measurements within the drone data, and by allometric equation.

**Table 1. Selected plot-aggregate aboveground carbon density (ACD) allometric equations for use with remotely-sensed height measurements.**

| Equation | Forest type | Sample data range | ACD equation | Reference |
|---|---|---|---|---|
| I | Pantropical forests | n plots = 754 | $ACD = 6.85 \times TCH^{0.952}$ | [57] |
| II | Lowland tropical rainforest, Sabah | n = 45,214; n plots = 36; DBH range: 12–165 cm; H range: 16–72 m | $ACD = 7.37 \times TCH^{0.87}$ | [58] |
| III | Lowland tropical rainforest, Sabah | n = 45,214; n plots = 36; DBH range: 12–165 cm; H range: 16–72 m | $ACD = 1.03 \times TCH^{1.535}$ | [58] |
| IV | Peat swamp pole forest, Kalimantan | n plots = 22 | $ACD = 0.47 \times TCH^{1.87}$ | [36] |
| V | Lowland tropical rainforest, Sabah | n = 261,937; n plots = 173 | $ACD = 0.567 \times TCH^{0.554} \times BA^{1.081} \times WD^{0.186}$ where $BA = 1.112 \times TCH$, $WD = 0.385 \times TCH^{0.097}$ | [24] |

ACD in Mg C ha$^{-1}$; TCH, mean top of canopy height in m; BA, stand basal area in m$^2$ ha$^{-1}$; WD, community-weighted mean wood density in g cm$^{-3}$. Forest types and underlying sample data ranges are given where available. H, crown height in m; DBH, diameter at breast height in cm.

### Aboveground carbon density measurements from field data

**Field data collection.** Field-based tree inventory data was collected for the 50 m x 50 m botanical plot (Fig 1) in October 2021. The team recorded the boundaries of the restoration site and the botanical plot using a Garmin GPSMAP 64s (± 3.7 m accuracy; Garmin, Olathe, USA). Diameter at breast height (DBH in cm) was measured for each tree (n = 24), as well as crown height (H in m) using a clinometer and tape measure. Wood density (WD in g cm$^{-3}$) was not directly measured, and field staff were unable to identify trees to the species or genus level. This meant that wood density estimates could not be obtained from species-specific databases, a common alternative to direct measurements in biomass studies [37]. Instead, we identified a range of plausible community mean WD values from published ecological studies of Southeast Asian rainforests [59–61], which informed the WD distributions used in the following error propagation steps.

**Error propagation in field measurements.** Adapting the workflow of Réjou-Méchain et al. [62], we propagated uncertainty in field-based measurements of DBH and H using the Monte Carlo method. To calculate uncertainty in WD, values were assigned from a normal distribution with a mean of 0.54 g cm$^{-3}$ and a standard deviation of 0.11 g cm$^{-3}$. Using the above terms, we ran 1,000 simulations for each tree within the plot (n = 24), resulting 1,000 sets of plot measurements.

**Individual tree allometric equations.** We used 27 different allometric equations to calculate the average ACD value for the botanical plot using the field data (S1 Table). Since most community organisations lack the capacity for direct sampling, we sought to understand the magnitude of over- or underestimation in ACD values derived from preexisting equations not calibrated with on-site sampling or based on different empirical datasets [37], necessitating a large selection of equations. We identified the 27 equations based on their applicability to the study site; they ranged in specificity from pantropical moist forests to individual forest reserves. All site-specific equations were derived from forests in Borneo or the neighbouring Indonesian island of Sumatra. As individual tree allometries calculate aboveground biomass (AGB in kg), plot-level AGB values were converted to ACD by combining the AGB values of all trees (n = 24) for each simulation, dividing by the plot area

(0.25 ha), and using a carbon content conversion factor of 0.47 [63]. This process resulted in a total of 27,000 ACD calculations for the botanical plot.

## Results

### Structure-from-motion outputs

The DSM and initial DEM produced from the point cloud had a final resolution of 0.1 m x 0.1 m. The DEM showed a large variation in elevation across the restoration site (21.4 m) and within the botanical plot (6.9 m; Fig 3). As mentioned previously, this variation did not correspond with the known elevation profile of the site (<1 m). Height variations were more pronounced towards the edge of the site and underneath denser vegetation and, though less prominent, also occurred in the botanical plot.

Canopy height values for the normalised CHM (corrected using a planar DEM; Fig 4) ranged from 0.38 m to 30.63 m. The mean TCH across the restoration site was 7.19 m ($\sigma$ = 6.19 m; median = 5.72 m). Canopy height within the botanical plot had a much smaller range, from 0.20 m to 22.60 m, with a mean TCH of 3.90 m ($\sigma$ = 4.41 m; median = 2.01 m).

Crown identification from our drone images required considerable field calibration. Fig 5A shows the locations of all tree crown tops >3 m found in the CHM by PyCrown, using estimate 5 (S1 Text) as an example. In Fig 5B, which focuses on the botanical plot, the grey lines indicate the delineated boundaries of the tree crowns found using the same parameters. The crown locations and extents identified in estimate 5 were generally accurate, albeit with some errors towards the edges of the restoration site. This pattern was typical of all five estimates. The field team identified 24 individual trees between 3 and 19 m high for analysis within the botanical plot. None of the five estimates produced using PyCrown returned the same number of tree crowns as the field team, with estimates ranging from 17 to 30 crowns. The crown heights derived from the drone data were similar to those measured in the field (Fig 6). The mean and median crown heights for the drone estimates ranged from 6.65 m to 8.25 m and 4.43 m to 5.81 m, respectively, while the field

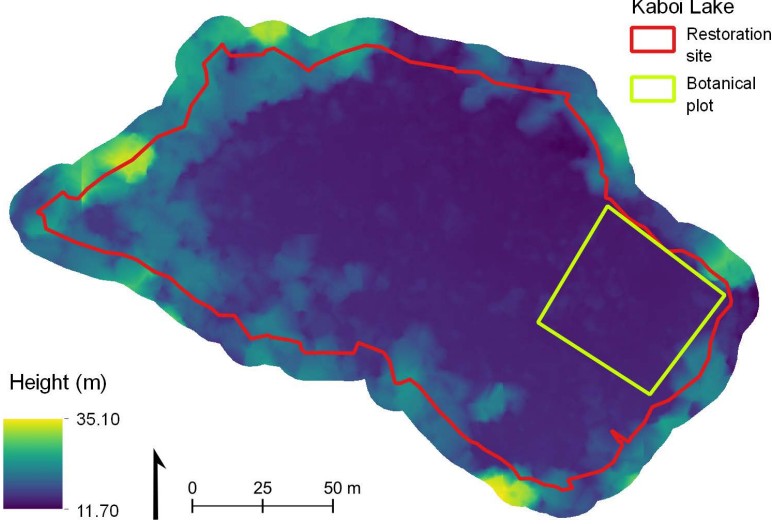

**Fig 3. Digital elevation model of the restoration site generated from classified point cloud.** 0.1 m resolution. Red line indicates the restoration site; green line indicates the botanical plot. Elevation is significantly higher towards the perimeter of the sire due to poor canopy penetration in the drone imagery.

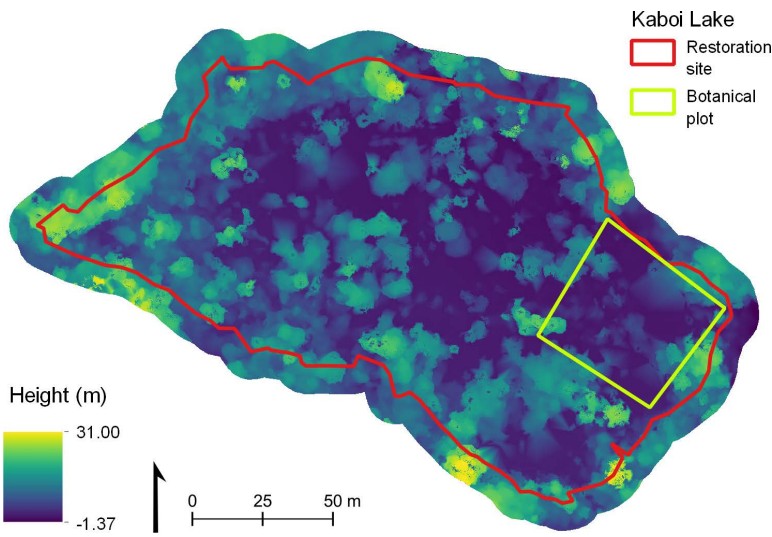

**Fig 4. Normalised canopy height model of the restoration site.** 0.1 m resolution. Red line indicates the restoration site; green line indicates the botanical plot. A flat, planar digital elevation model was used to normalise the point cloud-derived digital surface model.

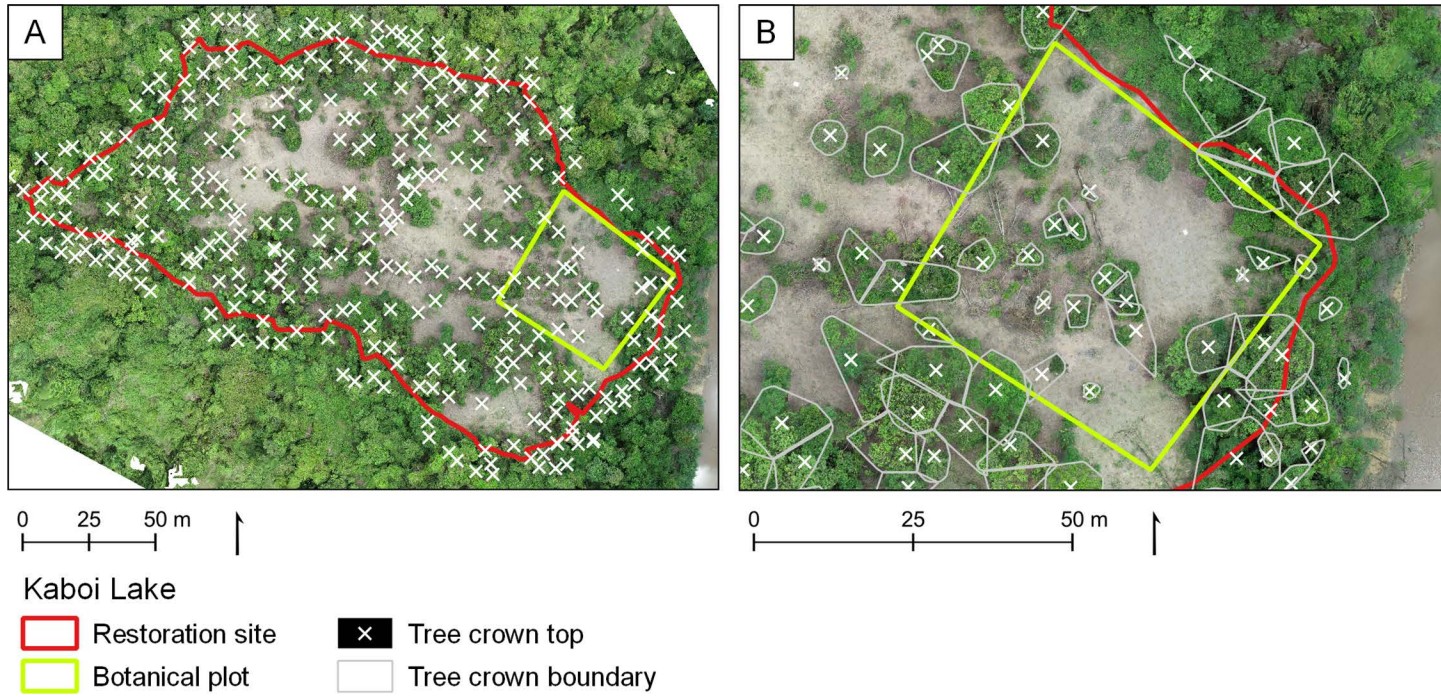

**Fig 5. Location and extent of tree crowns within the restoration site.** Tree crowns identified using PyCrown; figure shows results of PyCrown estimate 5. (A) Location of all tree crowns >3 m tall within the restoration site. (B) Location and extent of tree crowns >3 m tall within the botanical plot.

measurements had a mean height of 8.16 m and a median of 7.25 m. The drone estimates showed clear groupings of crowns <10 m, with fewer larger individuals. A similar pattern was observed in the field measurements, although with a greater number of crowns <13 m and only two crowns >15 m (Fig 6).

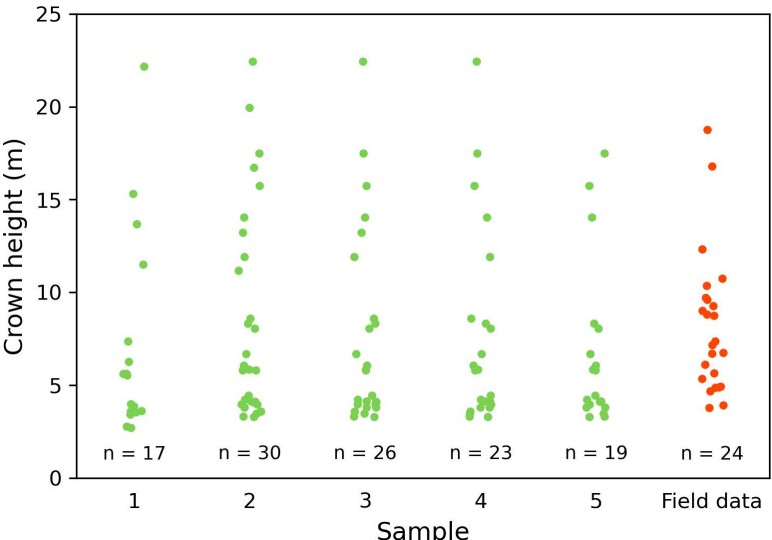

**Fig 6. Field- and drone-derived individual tree crown height measurements.** Samples 1–5 are measurements extracted from the canopy height model using different input parameters for PyCrown. Height measurements from field data shown in orange. Number of individual tree crowns >3 m identified by each sample is shown at the bottom.

## Aboveground carbon density measurements from drone data

Drone-derived estimates of biomass have significantly higher uncertainty compared to those based on field data. The distribution of ACD measurements for the botanical plot produced using five different plot-aggregate equations (Table 1) are shown in Fig 7. For comparison, Fig 7 also shows the combined distribution of all field-derived ACD measurements using 27 different allometric equations (S1 Table). Across all five drone-derived distributions, a fivefold variation in mean and median ACD values was observed. The ACD values calculated using the larger modelled height measurement errors ($\sigma$ = 4 m; Fig 7B) showed substantial differences in distribution ranges. The variation within the measurements for each equation was significantly greater with larger height measurement errors compared to the smaller errors ($\sigma$ = 1.5 m; Fig 7A).

With larger errors, the combined mean ACD value for all five equations was 16.78 Mg C ha$^{-1}$ ($\sigma$ = 17.79 Mg C ha$^{-1}$), compared to a field-derived mean ACD value of 6.05 Mg C ha$^{-1}$ ($\sigma$ = 2.07 Mg C ha$^{-1}$; all 27 equations) (Fig 7B). For smaller error estimates, the mean ACD was 14.06 Mg C ha$^{-1}$ ($\sigma$ = 10.64 Mg C ha$^{-1}$) (Fig 7A). There was a clear difference between the measurements produced by equations *I* and *II*, and equations *III-V*. Under both measurement error scenarios, equations *I* and *II* produced mean ACD values approximately four times higher than those derived from field data. The mean ACD values for equations *III-V* were lower, and those using smaller measurement errors more closely resembled field measurements. When equations *III-V* were combined, the mean ACD value was 7.19 Mg C ha$^{-1}$ ($\sigma$ = 4.68 Mg C ha$^{-1}$) with smaller errors, and 10.95 Mg C ha$^{-1}$ ($\sigma$ = 13.20 Mg C ha$^{-1}$) with larger errors. However, the range of ACD values for equations *III-V* exceeded that of the field measurements under both error distributions.

When applying the plot-aggregate equations across the whole restoration site and averaging the results, the carbon density value was twice that of the botanical plot. Using the smaller height error distribution, mean ACD was 29.28 Mg C ha$^{-1}$ ($\sigma$ = 13.61 Mg C ha$^{-1}$), and using large errors it was 31.27 Mg C ha$^{-1}$ ($\sigma$ = 22.63 Mg C ha$^{-1}$). When just equations *III-V* were

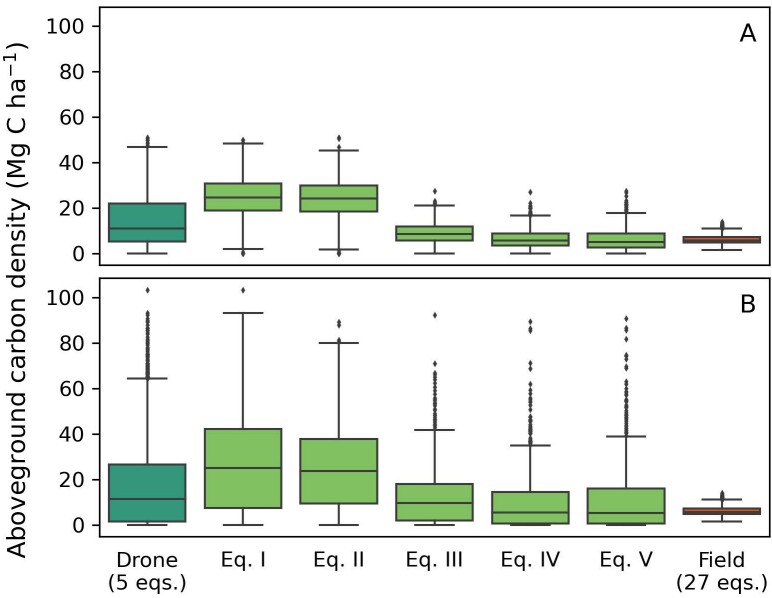

**Fig 7. Distributions of field- and drone-derived aboveground carbon density (ACD) values for the botanical plot.** For drone data, combined ACD values for all five allometric equations are shown in dark green, with individual equations in light green. For field data, combined ACD values from 27 allometric equations are shown in orange. **(A)** ACD distributions calculated using small-modelled errors in drone height measurements ($\sigma = 1.5$ m). **(B)** ACD distributions using large-modelled errors ($\sigma = 4$ m).

combined, mean ACD values were 20.24 Mg C ha$^{-1}$ ($\sigma = 7.50$ Mg C ha$^{-1}$) using small errors and 23.95 Mg C ha$^{-1}$ ($\sigma = 20.67$ Mg C ha$^{-1}$) using large errors.

## Discussion

### Aboveground carbon density measurements

Drone-based ACD calculations for our field plots were systematically higher than field-based measurements and had wider uncertainties (Fig 7). The mean drone-derived ACD measurements for the plot were approximately double the field-based carbon density, which we assume is a true-to-reality benchmark. Two commonly used pantropical allometric equations, equations 1 [64] and 20 [65] in S1 Table, frequently serve as 'general allometric equations' in individual tree AGB studies [37,66–68] or as the basis for new allometric models [24,58]. These equations produced ACD distributions either side of the mean field-derived ACD value from all 27 equations. This increased our confidence that the distribution of ACD values across the 27 equations represented a plausible range which contained the true ACD value for the plot making it suitable for comparison with the drone measurements.

Three of the drone-derived values (equations *III-V*) were more similar to the field-based values, albeit with greater variability. A key factor here is the underlying datasets for these equations: all were calibrated using field plots that share general geographical and ecological similarities with Kaboi Lake. In contrast, the generalised pantropical allometric equation *I* was developed using primarily Neo- and Afrotropical forest plots, which are structurally distinct from the forests of Borneo [69]. While equations *II* and *III* were both derived from Sepilok-Kabili Forest Reserve, equation *II* used sub-models to predict diameter at breast height and wood density, whilst *III* used field measurements. Both equations *I* and *II* overestimated carbon densities for the plot by a greater degree than regionally calibrated equations

*III-V*. These results indicate that the selection of allometric equations significantly influences the accuracy of ACD calculations from SfM data, with a generalised equation overestimating carbon density values by four times. However, drone-derived SfM can be a viable method for producing ACD values comparable to those of field-based methods at a community scale, provided the plot-aggregate allometric equations used were calibrated using ecologically and geographically appropriate datasets. Regionally-calibrated ITC allometric equations are readily available (e.g., S1 Table), but pre-published plot-aggregate equations are comparatively uncommon. The development of new regionally-calibrated plot-aggregate allometries for different ecoregions and species [e.g., 70–72] would greatly increase the applicability of this method for community use.

Differences in calculation methods and assumptions between the field- and drone-based approaches may explain the observed bias towards larger drone-derived ACD values. ITC approaches, like our field-based methods, calculate carbon within discrete units (individual trees), excluding smaller trees (those <3 m), low-lying vegetation, and deadfall from total carbon density calculations. In contrast, the plot-aggregate method used in this study did not differentiate between trees and non-trees, and included all biomass within the CHM when calculating mean TCH. While this theoretically results in higher carbon values, shorter trees and vegetation have a disproportionately small impact on total carbon in practice. Differences may also arise from large tree crowns that cross the plot boundary. These trees were not recorded by the field team as their trunks lay outside of the boundary but, due to the 'cookie cutter' methods used to extract values from the CHM, they did contribute to the overall carbon values calculated via the plot-aggregate approach. These edge effects were perhaps amplified by the small relative size of the plot [57]. Differences may also arise from uncertainties in the drone-derived CHM, which are discussed below.

Our calculated ACD values for Kaboi Lake are significantly lower than other published values for secondary forests in Borneo. Previously logged forests in Sabah can contain carbon densities of 60–140 Mg C ha$^{-1}$ [21], whilst for secondary peat forests in Kalimantan, ACD ranges from 40–100 Mg C ha$^{-1}$ [36,73]. These values are approximately an order of magnitude greater than those measured at the botanical plot. The low carbon density at Kaboi Lake could feasibly be explained by both the historic logging of dipterocarps and the recent clearing, and Asner et al. [21] show that recently deforested lands in Sabah (<5 years) have significantly lower carbon densities (7 Mg C ha$^{-1}$), more consistent with our results.

Differences between our results and other published ACD values for secondary forest suggest a potential for overestimation of baseline carbon density values at restoration sites, especially if using remotely sensed imagery with low resolution relative to site size. The drone-based methods we outline here offer a more accurate solution for assessing the baseline carbon values for community-scale ACD measurements compared to satellite-based methods. Further, the five plot-aggregated allometric equations (Table 1) were not necessarily developed and calibrated for use in severely degraded forest. The future use of drone SfM and plot-aggregate allometries specifically calibrated for severely degraded forest may reveal further differences between assumptions used in restoration planning and carbon accounting, and on-the-ground ACD values.

## Methodological limitations and uncertainties

Uncertainties in the drone-derived ACD values arise from both the selection of allometric models and generation of the CHM. Mean ACD measurements varied by a factor of 4 between equations using the smaller height measurement errors, and by a factor of 3 when using larger errors (Fig 7). Clear groupings emerged among the equations, with equations *III-V* more

closely matching field-derived measurements. This grouping is explained by the difference in underlying datasets used to produce the equations, highlighting the importance of equation selection for this method.

However, all individual plot-aggregate equations exhibited a much broader distribution of results compared to field measurements, reflecting the height measurement errors associated with drones. These broader distributions were caused by the size of the error distributions used to propagate uncertainty in the mean TCH values relative to the CHM height. The mean TCH value for the botanical plot was 3.9 m, while the error distributions had standard deviations of 1.5 m and 4 m. Using ground control points (GCPs) in the data collection phase could reduce the uncertainties surrounding drone height measurements [52,54], but Fig 7A shows that even with the reduced errors expected from GCP correction (i.e., modelled using the smaller error distribution), large uncertainties in ACD measurements remain.

The accuracy of the canopy height model is ultimately dependent on the digital surface and elevation models generated by SfM, with DEMs having a greater impact on accuracy due to the relative size of their measurement errors. Limited canopy penetration with optical imagery poses a challenge for SfM, resulting in fewer ground returns and poorer quality DEMs compared to LiDAR data [32,55,74–76]. Nevertheless, DEMs derived from optical drone imagery have been successfully used to measure forest biomass [31,77], especially in woodlands with relatively open canopies [78], similar to our study site. Although Kaboi Lake had visible bare ground, we achieved higher accuracy in our CHM by assuming a flat, low relief surface rather than using the DEM produced by SfM, which included a relief of 21.4 m. This approach is not feasible in regions of significant topographic relief or complex topography. Nevertheless, it avoids the issues of matching datasets from different sensors and platforms, making it a plausible technique for minimising errors in SfM-derived DEMs and CHMs, particularly when drone imagery is available from the pre-restoration forest clearance.

Ground control points (GCPs) are usually an important part of the SfM workflow, used to accurately locate, orient and scale point clouds in space [79]. However, we experienced technical issues in the acquisition and integration of GCPs into the ODM software. Hence, we analysed the data without ground controls and examined the impact of omitting this data collection process. We used only the drone's onboard global navigation satellite system (GNSS) receiver to provide geospatial data and scale the CHM, and used a comparison of tree heights from field measurements and the CHM to validate the scaling. The tree crown heights extracted using PyCrown followed similar distribution patterns to the field measurements, with the majority of individual crowns measuring <10 m across all measurements (Fig 6). However, clear differences emerged in the number of tree crowns identified in the botanical plot across PyCrown estimates. Increased numbers of taller trees (>10 m) identified within the plot may be explained by the presence of large, overhanging canopies from trees that are situated outside of the botanical plot.

The maximum field-measured crown height was 18.8 m, and omitting the (presumed overhanging) trees taller than 18.8 m from estimates 1–4 produces distributions more closely aligned with the field measurements but also reduces mean heights. The discrepancy in mean heights may be due to the downscaling of the CHM for PyCrown processing, which reduces the 'visibility' of fine-scale canopy peaks [80,81] and thereby reduces height measurements. The lower mean crown heights also follow other results showing a systematic underestimation of TCH using SfM [32,75,82,83]. However, additional studies have demonstrated SfM overestimating TCH in open canopy forest [81], or the bias shift changing with canopy height [84]. As this study utilised a flat DEM, it negated the impact of ground occlusion in the DEM which is often a major contributor to reported underestimations of canopy height. Of importance, then, is the fact that errors in field measurement methods were not considered in these

comparisons and are another potential source of bias. Canopy height is the key uncertainty in field measurements; DGFC staff estimated uncertainty in canopy height measurements at approximately 3 m, exacerbated by taller trees or the use of novice surveyors. Despite differences between the sets of measurements, the coincident uncertainties between field and drone-derived data suggest that the CHM was scaled sufficiently during the SfM process to enable plausible ACD measurements to be produced, as the uncertainties here were smaller than those associated with allometric equation selection.

## Implications of method for community-scale carbon monitoring

Our findings suggest that lightweight, low-cost, consumer-grade drones and open source software present a viable solution for generating ACD values within community-scale projects. There is an optimal scale for using drones for ACD measurements with regards to trade-offs between accuracy, simplicity, and cost-effectiveness. This optimal scale ranges between individual plot-level and regional-scale surveys, i.e., between approximately 1 and 100 ha. Between these bounds, drones offer an attractive option for data acquisition and carbon measurement, aligning well with the needs of community-scale ACD monitoring while bridging the gap between field-based and satellite-based measurements.

At scales between 1 and 100 ha, drone-derived ACD estimates can be obtained without extensive field surveys and using only a single input metric. Our findings further support the idea that drones offer a fast and cost-effective option for data acquisition at scales of up to tens of hectares [35,85,86]. A team of two people were able to map the entire 2 ha restoration plot at a high resolution (5 cm) in a single morning, whereas collecting field-based measurements for each tree in the same plot would take two people several days. Due to the reduction in survey time per unit area surveyed, the drone-based method we demonstrate here is a promising option for scaling up carbon monitoring from a botanical plot level. For example, canopy height metrics for a 10 ha site can be measured using drones more quickly than gathering field measurements for a single 0.25–1 ha plot. While field plots remain necessary for calibration and verification, this approach significantly reduces total survey times.

However, at smaller scales (<2 ha) and with one-off surveys, it is worth recognising that it may be simpler, faster, and cheaper to utilise field-based methods over drone-based SfM. Although field-based methods do require more input metrics and require certain surveying skills, they do not require training in piloting and data processing, nor the purchase of comparatively expensive hardware – the drone used here cost approximately £1,500 (field staff already had access to a smartphone for mission planning). Still, with larger areas or repeat surveys, the simplicity and potential accuracy benefits of field-based methods may be outweighed by the subsequent financial advantages (e.g., reduced labour costs) of drone-based SfM.

Drone use encounters practical limitations at larger scales. The high temporal and spatial resolution of drone imagery allows for better detection of forest structure than freely available imagery that could be used for larger-scale (>100 ha) ACD measurements (e.g., Landsat or ESA's CCI biomass dataset). Whilst drone-based SfM has been used over these scales [32], there are potential trade-offs between resolution, extent and labour costs (greater spatial resolution may require more, lower altitude flights). The relatively short range of drones also introduces issues concerning travelling to launch sites, both in terms of accessibility and total survey times. For surveys >100ha, purchasing high-resolution (30 cm) snapshot satellite imagery for a site, or even commissioning an airborne LiDAR survey, may become a more practical option (e.g., a WorldView-3 satellite image encompassing the site would have cost ≈£400). These approaches do, however, come with disadvantages related to temporal resolution and repeatability, and would still require field-based measurements of ACD within botanical plots to calibrate imagery.

Access to drones and drone imagery also provides secondary benefits for restoration projects and forest communities alongside community-scale ACD monitoring. Orthomosaic images are an effective and transparent way of demonstrating tree planting and restoration progress, a task that is difficult with lower spatial or temporal resolution imagery. Although numbers of trees planted is not necessarily a strong measure of restoration success [18], it can be an important metric for funding partners. Drones can capture compelling images of a site and its surrounding landscape for use in social media and outreach campaigns run by restoration projects. In Borneo, some communities have used these images to create postcards and calendars to sell locally and to promote restoration projects as tourist attractions, providing additional sources of revenue [87]. Beyond restoration, the georeferenced maps produced from drone imagery can also be used to assert land rights and stop extractive industries from operating within community-owned forest [87,88].

Community groups often have limited technical and financial resources, making low-cost, accessible methods like the one presented here especially valuable for community-scale carbon monitoring. Nevertheless, there are several factors that may limit this method's accessibility for community use. First is the need for, access to, and costs of pilot training. Piloting a multirotor drone may be straightforward, but precise flight planning is required to maximise the accuracy of any SfM outputs. Variables such as sun angle during image capture, camera angle, and image overlap significantly affect point cloud construction [35,89]. A few days of training should be sufficient to pilot a multirotor safely, set up and record GCPs, and collect imagery suitable for SfM, though more training may be needed for fixed wing drones.

Second is the role of data processing; it is easy to focus on flying a drone, but this is only half the process of producing ACD measurements. Any community-scale groups or actors wishing to replicate these methods will need a good working knowledge of GIS, Python and relevant open source software, such as ODM. This, again, may require additional training but open source programs are increasingly packaged with accessible, user-friendly interfaces alongside more technical command line options. Data processing also takes a considerable time; processing ~600 images and producing point clouds took over 3 hours on a powerful desktop PC. Added to this are the multiple attempts over several days that failed part way through due to insufficient memory. Using lower-resolution imagery reduces processing times, although in our experience this results in greater measurement errors due to ground occlusion and image matching issues [ cf. 90]. In combination, lengthy data processing steps may further reduce time advantages over manual field sampling (albeit less so for larger sites).

Finally, there are considerable additional expenses beyond just purchasing a drone. A laptop capable of running the SfM and data processing software may cost as much as the drone itself (up to approximately £1,000). However, like a drone, its applicability for other purposes may counterbalance these additional costs. A tablet is required to operate the drone, although smartphones, which can also be used, are becoming increasingly common even in rural areas. Surveys with consumer-grade drones often require additional hardware, such as handheld GNSS receivers for recording GCPs (≈£300 for a basic unit), and paid subscriptions to photogrammetry software (PIX4Dmapper, a popular photogrammetry program, currently costs ≈£220 per month; www.pix4d.com). As demonstrated in this study, open source photogrammetry software can reduce costs, as can forgoing GCPs and using geolocation data embedded in the input images. Additionally, there are the costs associated with obtaining permits or certificates required to fly in the region. The costs here may be small, but the legislation introduces an additional potential barrier, as community groups may find navigating the myriad forms and administrative requirements more difficult than academics with connections to local universities and forestry departments.

One solution to overcoming these obstacles is for communities to partner with NGOs and research institutes to help with drone operations. For example, in Indonesia, Swandiri Institute are one of a handful of organisations providing community drone training and capacity building, while others like the Center for International Forestry Research (CIFOR) can conduct data collection and processing on a community's behalf. Private organisations can also provide this service for a fee, which may be a cost-effective alternative to purchasing a drone, training courses, and permits for one-off surveys. However, such 'drone outsourcing' [87] can risk entrusting key ethical decisions around consent, privacy, data ownership, and the handling of potentially incriminating images to the contracted party, with potential negative impacts for the local community [91,92]. Outsourcing also restricts working knowledge of drones and data processing to a smaller number of individuals in a region. In situations where communities are proactive participants in drone mapping with NGO partners, they are still often dependent on NGOs for technical expertise [87,93–95]. Building local capacity is an important factor in increasing the long-term sustainability of community-based drone monitoring and reducing potentially negative impacts.

Barriers to accessibility do not only apply to the use of drones for carbon monitoring, nor are they geographically limited to Borneo. Drones will always interact with real-world factors that can limit the accessibility of such methods. Conservation spaces differ significantly from controlled environments like testing laboratories or university campuses and can present unexpected challenges [96]. In our case, extreme temperatures limited the duration of drone surveys, whilst routine flooding delayed data collection for several months. It is worth considering how these environmental factors might affect the practical use of other conservation and remote sensing technologies. Additionally, factors like species identification skills or data-handling capacities may limit other participatory monitoring approaches, even when drones are not involved. Awareness of these factors is important for managing expectations around new remote sensing technologies and for making methodologies accessible and relevant to those who will benefit from them most, not only in Borneo, but in forest ecosystems and conservation spaces in general.

## Conclusions

In this paper, we developed, applied, and analysed a new method for incorporating consumer-grade drones into community-scale aboveground carbon measurements, utilising open source software, drone-derived SfM, and pre-published plot-aggregate allometric equations. Our results show that this method presents a viable option for generating ACD measurements for community-scale conservation and restoration projects, producing results comparable to those obtained using established field-based methods. Drone-derived measurements were larger than field-derived measurements, but varied depending on the allometric models used. This highlights the importance of selecting regionally calibrated allometric equations when applying this method. The development of new models for a range of forest types across the tropics will greatly increase this method's accuracy and applicability.

The approach presented here offers several advantages over existing methodologies that could be used for community-scale ACD measurements, including a reduction in survey times and long-term costs. However, several factors may limit the accessibility of this method for community groups in practice. These barriers – analogous to those in other methodologies, technologies, and locations – may be resolved with relative ease, but should not be overlooked. Nevertheless, the method described here has established a foundation for a simple drone-based workflow to measure carbon, showing promise for real-world applications and potential refinement in future studies.

## Supporting information

**S1 Text. Processing parameters for tree crown location estimates generated using PyCrown.**
(DOCX)

**S1 Table. Selected allometric equations used to generate aboveground biomass (AGB) distributions from field-derived measurements.** AGB in kg; DBH, diameter at breast height in cm; H, tree height in m; WD, wood density in g cm³; W, weight in kg. Forest types and underlying sample data ranges are given where available.
(DOCX)

## Acknowledgements

We are grateful to the Sabah Biodiversity Centre, the Sabah Forestry Department, and the Sabah Wildlife Department for permissions to carry out the research and for collaboration with the project. We also wish to thank KOPEL Bhd, Regrow Borneo, and research assistants at the Danau Girang Field Centre for their efforts in collecting field data and their activities at the restoration site.

## Author contributions

**Conceptualization:** Ben Newport.

**Formal analysis:** Ben Newport.

**Funding acquisition:** Ben Newport.

**Investigation:** Amaziasizamoria Jumail.

**Methodology:** Ben Newport.

**Visualization:** Ben Newport.

**Writing – original draft:** Ben Newport.

**Writing – review & editing:** Ben Newport, Tristram C. Hales, Joanna House, Benoit Goossens.

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
