## [Decision Letter · Decision Letter 0]

18 Nov 2024

PONE-D-24-34507Simplifying drone-based aboveground carbon density measurements to support community forestryPLOS ONE

Dear Dr. Newport,

Thank you for submitting your manuscript to PLOS ONE. After careful consideration, we feel that it has merit but does not fully meet PLOS ONE’s publication criteria as it currently stands. Therefore, we invite you to submit a revised version of the manuscript that addresses the points raised during the review process.

The manuscript presents a compelling and timely study on the integration of low-cost drones into community-scale carbon monitoring projects. Both reviewers provided insightful feedback that highlights the strengths of the manuscript while identifying areas for improvement. Below, I summarize their main points and offer guidance for revisions.

Both reviewers commend the relevance of the study, particularly in addressing an important gap in carbon monitoring by focusing on accessible methods for community forestry projects. To further strengthen the manuscript, consider emphasizing the broader implications of community forestry in the introduction and discussion sections. Highlighting how community-based approaches bridge conservation and local economic needs will enhance the study's impact. Reviewer 1 noted the importance of regionally calibrated allometric models and suggested elaborating on how these influence the accuracy of carbon measurements. Expanding on this point, particularly in the discussion, could provide valuable insights for readers. Additionally, consider discussing the potential for developing models tailored to different tropical forest types, as this could further enhance the applicability of the method. Reviewer 1 appreciated the discussion of barriers to community adoption of the proposed method but suggested including practical solutions, such as training programs or partnerships. Integrating specific, actionable recommendations in this section would enhance the manuscript's utility for community groups and practitioners. Reviewer 2 also highlighted the importance of transparency in addressing methodological challenges, which aligns well with this theme. Both reviewers acknowledge the thorough discussion of data processing and costs. However, Reviewer 1 suggested adding more details on optimizing these steps, particularly for communities with limited resources. Including examples of low-cost or open-source software and practical strategies for data processing could address this suggestion effectively. Reviewer 1 identified areas in the discussion section where the writing could be reorganized to improve clarity and flow. Consider revisiting this section to enhance the presentation of methodological limitations and their implications. Reviewer 1 mentioned providing specific suggestions in the manuscript, so these could serve as a helpful starting point for revisions. Reviewer 2 requested the inclusion of the variation in the mean and median for "heights of the crown" data obtained in the field (line 282) to match the reporting for drone-derived data in line 281. Including this information will improve consistency and clarity in the data presentation. Reviewer 2 noted that the resolution of all images needs improvement. Ensuring high-quality figures will enhance the overall readability and visual appeal of the manuscript.

In summary, the manuscript is a valuable contribution to the field, and with minor revisions addressing the points above, it will provide even greater technical, social, and ecological insights. By incorporating the reviewers' suggestions, the study will not only advance carbon monitoring methodologies but also underscore the vital role of community forestry in achieving sustainable development goals.

We look forward to receiving your revised manuscript.

Kind regards,

Luisa Maria Diele-Viegas, M.D.

Academic Editor

PLOS ONE

Journal Requirements:

2. We note that you have indicated that there are restrictions to data sharing for this study. PLOS only allows data to be available upon request if there are legal or ethical restrictions on sharing data publicly. For more information on unacceptable data access restrictions, please see http://journals.plos.org/plosone/s/data-availability#loc-unacceptable-data-access-restrictions. Before we proceed with your manuscript, please address the following prompts: a) If there are ethical or legal restrictions on sharing a de-identified data set, please explain them in detail (e.g., data contain potentially identifying or sensitive patient information, data are owned by a third-party organization, etc.) and who has imposed them (e.g., a Research Ethics Committee or Institutional Review Board, etc.). Please also provide contact information for a data access committee, ethics committee, or other institutional body to which data requests may be sent. b) If there are no restrictions, please upload the minimal anonymized data set necessary to replicate your study findings to a stable, public repository and provide us with the relevant URLs, DOIs, or accession numbers. For a list of recommended repositories, please see https://journals.plos.org/plosone/s/recommended-repositories. You also have the option of uploading the data as Supporting Information files, but we would recommend depositing data directly to a data repository if possible. We will update your Data Availability statement on your behalf to reflect the information you provide.

3. We note that Figure 1 and 5 in your submission contain map/satellite images which may be copyrighted. All PLOS content is published under the Creative Commons Attribution License (CC BY 4.0), which means that the manuscript, images, and Supporting Information files will be freely available online, and any third party is permitted to access, download, copy, distribute, and use these materials in any way, even commercially, with proper attribution. For these reasons, we cannot publish previously copyrighted maps or satellite images created using proprietary data, such as Google software (Google Maps, Street View, and Earth). For more information, see our copyright guidelines: http://journals.plos.org/plosone/s/licenses-and-copyright. We require you to either (1) present written permission from the copyright holder to publish these figures specifically under the CC BY 4.0 license, or (2) remove the figures from your submission:

a. You may seek permission from the original copyright holder of Figure 1 and 5  to publish the content specifically under the CC BY 4.0 license. We recommend that you contact the original copyright holder with the Content Permission Form (http://journals.plos.org/plosone/s/file?id=7c09/content-permission-form.pdf) and the following text: “I request permission for the open-access journal PLOS ONE to publish XXX under the Creative Commons Attribution License (CCAL) CC BY 4.0 (http://creativecommons.org/licenses/by/4.0/). Please be aware that this license allows unrestricted use and distribution, even commercially, by third parties. Please reply and provide explicit written permission to publish XXX under a CC BY license and complete the attached form.” Please upload the completed Content Permission Form or other proof of granted permissions as an "Other" file with your submission. In the figure caption of the copyrighted figure, please include the following text: “Reprinted from [ref] under a CC BY license, with permission from [name of publisher], original copyright [original copyright year].”

b. If you are unable to obtain permission from the original copyright holder to publish these figures under the CC BY 4.0 license or if the copyright holder’s requirements are incompatible with the CC BY 4.0 license, please either i) remove the figure or ii) supply a replacement figure that complies with the CC BY 4.0 license. Please check copyright information on all replacement figures and update the figure caption with source information. If applicable, please specify in the figure caption text when a figure is similar but not identical to the original image and is therefore for illustrative purposes only. The following resources for replacing copyrighted map figures may be helpful: USGS National Map Viewer (public domain): http://viewer.nationalmap.gov/viewer/ The Gateway to Astronaut Photography of Earth (public domain): http://eol.jsc.nasa.gov/sseop/clickmap/ Maps at the CIA (public domain): https://www.cia.gov/library/publications/the-world-factbook/index.html and https://www.cia.gov/library/publications/cia-maps-publications/index.html NASA Earth Observatory (public domain): http://earthobservatory.nasa.gov/ Landsat: http://landsat.visibleearth.nasa.gov/ USGS EROS (Earth Resources Observatory and Science (EROS) Center) (public domain): http://eros.usgs.gov/# Natural Earth (public domain): http://www.naturalearthdata.com/

Reviewers' comments:

Reviewer's Responses to Questions

**Comments to the Author**

1. Is the manuscript technically sound, and do the data support the conclusions?

Reviewer #1: Yes

Reviewer #2: Yes

2. Has the statistical analysis been performed appropriately and rigorously? 

Reviewer #1: Yes

Reviewer #2: Yes

3. Have the authors made all data underlying the findings in their manuscript fully available?

Reviewer #1: Yes

Reviewer #2: Yes

4. Is the manuscript presented in an intelligible fashion and written in standard English?

Reviewer #1: Yes

Reviewer #2: Yes

5. Review Comments to the Author

Reviewer #1: Overall, I really liked the idea of the article and the approach adopted by the authors. The study presents an interesting and relevant proposal for incorporating low-cost drones into community-scale carbon monitoring projects. Additionally, the care taken with the analyses is evident, with a detailed description of the study area and the methods employed, which contributes to the robustness of the study.

- The authors clearly presented the variability between drone-derived carbon measurements and field-based ones. I find it valuable that they pointed out the importance of choosing regionally calibrated allometric models, but it would be helpful to expand a bit more on how this influences measurement accuracy. Furthermore, developing more specific models for different types of tropical forests could indeed improve the accuracy and applicability of this approach.

- One aspect that stood out was the authors' consideration of the possible barriers to applying this method by community groups. While well-founded, I believe the article could benefit from suggesting practical solutions, such as training programs or partnerships that could minimize these challenges, making the method more accessible.

- The processing of data and associated costs were well-addressed in the text. However, it would be interesting to include more details on how to optimize these steps, especially for community groups that may not have access to high-performance equipment or software.

- In general, the writing is clear, but some sections, especially in the discussion, could be improved for better understanding. I would suggest reorganizing some parts to enhance the flow of ideas, particularly when discussing methodological limitations and how they impact the results. I have included some specific suggestions in the manuscript to simplify the text and make certain points easier to follow.

- One of the strengths of this manuscript is its focus on community-scale forestry projects and the role they play in both carbon sequestration and sustainable development. However, I believe the manuscript would benefit from emphasizing the broader relevance of community forestry both in the introduction and the discussion. Highlighting the unique role that community forestry plays in bridging conservation goals with local economic needs would make the study's rationale even stronger and more impactful, helping the reader understand the broader context of the study from the outset. In the discussion, the relevance of this method for community forestry could be further elaborated. Community groups often lack access to costly technology and advanced scientific methods, making low-cost, accessible approaches like the one presented in this manuscript especially valuable. This addition highlights the social and environmental impact of community forestry and reinforces the importance of making the research accessible and relevant to those who will benefit most from it.

In summary, with minor adjustments to improve the method’s accessibility, as well as more detailed consideration of the practical challenges and benefits for communities, this work can make a strong contribution to the field. By addressing these points, the manuscript will not only advance the technical understanding of carbon monitoring but also support the social and ecological goals of community-based forest restoration.

Reviewer #2: The article deals with an important theme, because it brings a viable alternative for storage of carbon in biomass above ground, in specific situations as in the case discussed in the article.

The difficulties, limitations and adjustments of errors applied in the method and also possible biases were addressed which shows transparency in the process of data collection and treatment.

Please put in line 282 the variation of the mean and median of the variable "heights of the crown" (data obtained in the field), as it did in line 281 (for the data obtained with the drone).

The resolution of all images needs to be improved.

6. PLOS authors have the option to publish the peer review history of their article (what does this mean? ). If published, this will include your full peer review and any attached files.

**Do you want your identity to be public for this peer review?** For information about this choice, including consent withdrawal, please see our Privacy Policy .

Reviewer #1: No

Reviewer #2: No

---

## [Author Response · Author response to Decision Letter 1]

11 Feb 2025

Dear editors and reviewers,

Many thanks for considering our manuscript and sending us your comments and suggestions. Below we have listed the individual reviewer comments and the ways in which we have addressed them.

Reviewer 1

1. The authors clearly presented the variability between drone-derived carbon measurements and field-based ones. I find it valuable that they pointed out the importance of choosing regionally calibrated allometric models, but it would be helpful to expand a bit more on how this influences measurement accuracy. Furthermore, developing more specific models for different types of tropical forests could indeed improve the accuracy and applicability of this approach.

a. Line 354 – Added additional detail on the overestimation of ACD by generalised equations

b. Line 358 – Highlighted the difference in availability between individual tree and plot-aggregate equations, and how the development of more regional plot-aggregate equations would increase our method’s applicability.

2. One aspect that stood out was the authors' consideration of the possible barriers to applying this method by community groups. While well-founded, I believe the article could benefit from suggesting practical solutions, such as training programs or partnerships that could minimize these challenges, making the method more accessible.

a. Line 536 – Added a new paragraph to introduce options such as training programs and outsourcing data collection, alongside some of the potential drawbacks to these approaches that should be considered.

3. The processing of data and associated costs were well-addressed in the text. However, it would be interesting to include more details on how to optimize these steps, especially for community groups that may not have access to high-performance equipment or software.

a. Line 513 – Added nuance to the difficulties with open source programs.

b. Line 517 – Suggested one possible solution to lowering processing times with caveats.

c. Line 536 – The additional paragraph on Line 536 is perhaps also relevant for addressing this comment.

4. In general, the writing is clear, but some sections, especially in the discussion, could be improved for better understanding. I would suggest reorganizing some parts to enhance the flow of ideas, particularly when discussing methodological limitations and how they impact the results. I have included some specific suggestions in the manuscript to simplify the text and make certain points easier to follow.

a. We have incorporated many of the specific suggestions that were added to the manuscript, including in-line comments (e.g., defining allometric equations; Line 63).

5. One of the strengths of this manuscript is its focus on community-scale forestry projects and the role they play in both carbon sequestration and sustainable development. However, I believe the manuscript would benefit from emphasizing the broader relevance of community forestry both in the introduction and the discussion. Highlighting the unique role that community forestry plays in bridging conservation goals with local economic needs would make the study's rationale even stronger and more impactful, helping the reader understand the broader context of the study from the outset. In the discussion, the relevance of this method for community forestry could be further elaborated. Community groups often lack access to costly technology and advanced scientific methods, making low-cost, accessible approaches like the one presented in this manuscript especially valuable. This addition highlights the social and environmental impact of community forestry and reinforces the importance of making the research accessible and relevant to those who will benefit most from it.

a. Lines 45 and 49 – Added additional sentences on the co-benefits of community forestry for communities and conservation alike, as well as the long-term advantages of community forestry over industrial planting/restoration projects.

b. Line 490 – Added a new paragraph outlining some of the secondary benefits of drones and drone data for communities who use/access them for conservation purposes. This reinforces the importance of making drones and related technologies more accessible for community groups, and the impact that scientific research and methodologies can have beyond an academic context.

c. Line 501 – Changed some sentences to echo the points raised in this comment.

Reviewer 2

1. Please put in line 282 the variation of the mean and median of the variable "heights of the crown" (data obtained in the field), as it did in line 281 (for the data obtained with the drone).

a. Unfortunately we did not quite understand this comment: we have not given the variation of the mean and median for the field measurements as it was only a single distribution. The drone measurements on Line 281 have a range of values as we are summarising 6 separate distributions.

2. The resolution of all images needs to be improved.

a. The images we uploaded were the correct resolution and scale according to the PLOS ONE guidelines. We notice, however, that the previews included in the document “PONE-D-24-34507_reviewer.docx” are of a much lower resolution. Hopefully this will not be the case for the final publication.

Additional requirements

1. Please ensure that your manuscript meets PLOS ONE’s style requirements, including those for file naming.

a. This has been addressed.

2. We note that you have indicated that there are restrictions to data sharing for this study. PLOS only allows data to be available upon request if there are legal or ethical restrictions on sharing data publicly. … b) If there are no restrictions, please upload the minimal anonymized data set necessary to replicate your study findings to a stable, public repository and provide us with the relevant URLs, DOIs, or accession numbers.

a. Our data has now been uploaded to the CEDA Archive, a NERC repository for earth observation data.

b. The catalogue record link is: https://catalogue.ceda.ac.uk/uuid/98692ec457ee431cacc4027820e46411/

c. The DOI (when it is issued) will be: https://dx.doi.org/10.5285/98692ec457ee431cacc4027820e46411/

3. We note that Figure 1 and 5 in your submission contain map/satellite images which may be copyrighted. … We require you to either (1) present written permission from the copyright holder to publish these figures specifically under the CC BY 4.0 license, or (2) remove the figures from your submission.

a. The orthomosaic maps in Figures 1 and 5 were created by Author 1, using images that were collected by the DGFC team with a drone and used for the main analysis of this paper. We therefore do not believe that there are any copyright issues with those figures.

a. Reference list has been reviewed and the following references have been changed:

1. Tyukavina A, Baccini A, Hansen MC, Potapov P V., Stehman S V., Houghton RA, et al. Corrigendum: Aboveground carbon loss in natural and managed tropical forests from 2000 to 2012 (2015 Environ. Res. Lett. 10 074002). Environ Res Lett. 2018;13: 109501. doi:10.1088/1748-9326/aae31e

4. Olsson L, Barbosa H, Bhadwal S, Cowie A, Delusca K, Flores-Renteria D, et al. Land degradation. In: Shukla PR, Skea J, Calvo Buendia E, Masson-Delmotte V, Pörtner H-O, Roberts DC, et al., editors. Climate Change and Land: an IPCC special report on climate change, desertification, land degradation, sustainable land management, food security, and greenhouse gas fluxes in terrestrial ecosystems. Cambridge University Press; 2019. pp. 345–436. doi:10.1017/9781009157988.006

28. Cranston KA, Wong WY, Knowlton S, Bennett C, Rivadeneira S. Five psychological principles of codesigning conservation with (not for) communities. Zoo Biol. 2022;41: 409–417. doi:10.1002/zoo.21725

31. Otero V, Van De Kerchove R, Satyanarayana B, Martínez-Espinosa C, Fisol MA Bin, Ibrahim MR Bin, et al. Managing mangrove forests from the sky: Forest inventory using field data and Unmanned Aerial Vehicle (UAV) imagery in the Matang Mangrove Forest Reserve, peninsular Malaysia. For Ecol Manage. 2018;411: 35–45. doi:10.1016/j.foreco.2017.12.049

48. Zörner J, Dymond J, Shepherd J, Jolly B. PyCrown - Fast raster-based individual tree segementation for LiDAR data. Landcare Research NZ Ltd; 2018. doi:10.7931/M0SR-DN55

66. Kitayama K, editor. Co-benefits of Sustainable Forestry. Tokyo: Springer Japan; 2013. doi:10.1007/978-4-431-54141-7

b. These new references have been added:

9. Cranston KA. Building & Measuring Psychological Capacity for Biodiversity Conservation. Antioch University, New England. 2016. Available: https://aura.antioch.edu/etds/293

12. Duguma L, Minang P, Aynekulu E, Carsan S, Nzyoka J, Bah A, et al. From Tree Planting to Tree Growing: Rethinking Ecosystem Restoration Through Trees. Nairobi; 2020. Report No.: 304. doi:10.5716/WP20001.PDF

13. Fox H, Cundill G. Towards Increased Community-Engaged Ecological Restoration: A Review of Current Practice and Future Directions. Ecol Restor. 2018;36: 208–218. doi:10.3368/er.36.3.208

14. Kodikara KAS, Mukherjee N, Jayatissa LP, Dahdouh-Guebas F, Koedam N. Have mangrove restoration projects worked? An in-depth study in Sri Lanka. Restor Ecol. 2017;25: 705–716. doi:10.1111/rec.12492

15. Martin MP, Woodbury DJ, Doroski DA, Nagele E, Storace M, Cook-Patton SC, et al. People plant trees for utility more often than for biodiversity or carbon. Biol Conserv. 2021;261: 109224. doi:10.1016/j.biocon.2021.109224

16. Coleman EA, Schultz B, Ramprasad V, Fischer H, Rana P, Filippi AM, et al. Limited effects of tree planting on forest canopy cover and rural livelihoods in Northern India. Nat Sustain. 2021;4: 997–1004. doi:10.1038/s41893-021-00761-z

17. Lewis SL, Wheeler CE, Mitchard ETA, Koch A. Restoring natural forests is the best way to remove atmospheric carbon. Nature. 2019;568: 25–28. doi:10.1038/d41586-019-01026-8

70. Hao H, Li W, Zhao X, Chang Q, Zhao P. Estimating the Aboveground Carbon Density of Coniferous Forests by Combining Airborne LiDAR and Allometry Models at Plot Level. Front Plant Sci. 2019;10: 917. doi:10.3389/fpls.2019.00917

71. Cushman KC, Burley JT, Imbach B, Saatchi SS, Silva CE, Vargas O, et al. Impact of a tropical forest blowdown on aboveground carbon balance. Sci Rep. 2021;11: 11279. doi:10.1038/s41598-021-90576-x

72. Nunes M, Ewers R, Turner E, Coomes D. Mapping Aboveground Carbon in Oil Palm Plantations Using LiDAR: A Comparison of Tree-Centric versus Area-Based Approaches. Remote Sens. 2017;9: 816. doi:10.3390/rs9080816

87. Newport B. Going vertical: Exploring the technical opportunities and socio-political dynamics of drones in forest conservation. University of Bristol. 2024. Available: https://research-information.bris.ac.uk/en/studentTheses/going-vertical

88. Radjawali I, Pye O, Flitner M. Recognition through reconnaissance? Using drones for counter-mapping in Indonesia. J Peasant Stud. 2017;44: 817–833. doi:10.1080/03066150.2016.1264937

90. Gülci S. The determination of some stand parameters using SfM-based spatial 3D point cloud in forestry studies: an analysis of data production in pure coniferous young forest stands. Environ Monit Assess. 2019;191: 495. doi:10.1007/s10661-019-7628-4

91. Sandbrook C, Luque-Lora R, Adams WM. Human Bycatch: Conservation Surveillance and the Social Implications of Camera Traps. Conserv Soc. 2018;16: 493–504. doi:10.4103/cs.cs_17_165

92. Millner N, Newport B, Sandbrook C, Simlai T. Between monitoring and surveillance: Geographies of emerging drone technologies in contemporary conservation. Prog Environ Geogr. 2024;3: 17–39. doi:10.1177/27539687241229739

93. Cummings AR, Cummings GR, Hamer E, Moses P, Norman Z, Captain V, et al. Developing a UAV-Based Monitoring Program with Indigenous Peoples. J Unmanned Veh Syst. 2017; 115–125. doi:10.1139/juvs-2016-0022

94. Sauls LA, Paneque-Gálvez J, Amador-Jiménez M, Vargas-Ramírez N, Laumonier Y. Drones, communities and nature: pitfalls and possibilities for conservation and territorial rights. Glob Soc Challenges J. 2023;2: 24–46. doi:10.1332/AJHA9183

95. Vargas-Ramírez N, Paneque-Gálvez J. The Global Emergence of Community Drones (2012–2017). Drones. 2019;3: 76. doi:10.3390/drones3040076

We would like to thank both reviewers for taking the time to engage with our manuscript and for supporting what we hope is an improved final article.

---

## [Decision Letter · Decision Letter 1]

18 Mar 2025

Simplifying drone-based aboveground carbon density measurements to support community forestry

PONE-D-24-34507R1

Dear Dr. Newport,

We’re pleased to inform you that your manuscript has been judged scientifically suitable for publication and will be formally accepted for publication once it meets all outstanding technical requirements.

Kind regards,

Luisa Maria Diele-Viegas, M.D.

Academic Editor

PLOS ONE

Reviewers' comments:

Reviewer's Responses to Questions

**Comments to the Author**

1. If the authors have adequately addressed your comments raised in a previous round of review and you feel that this manuscript is now acceptable for publication, you may indicate that here to bypass the “Comments to the Author” section, enter your conflict of interest statement in the “Confidential to Editor” section, and submit your "Accept" recommendation.

Reviewer #1: All comments have been addressed

2. Is the manuscript technically sound, and do the data support the conclusions?

Reviewer #1: Yes

3. Has the statistical analysis been performed appropriately and rigorously? 

Reviewer #1: Yes

4. Have the authors made all data underlying the findings in their manuscript fully available?

Reviewer #1: Yes

5. Is the manuscript presented in an intelligible fashion and written in standard English?

Reviewer #1: Yes

6. Review Comments to the Author

Reviewer #1: Dear Authors,

The revisions really improved the manuscript and addressed all the key points from the first review. The expanded discussion on regionally calibrated allometric models (lines 354, 358) makes the method more accurate and applicable, and the added practical solutions, like training programs and outsourcing data collection (line 536), help make it more accessible for communities. The refinements in data processing and cost optimization (lines 513, 517) also add useful insights, especially for those working with limited resources.

The adjustments in the discussion sections improved the clarity and flow, making the study easier to follow. The stronger focus on community forestry (lines 45, 49, 490) does a great job of showing the broader social and environmental impact of this approach. Overall, the manuscript is much more solid and well-rounded.

7. PLOS authors have the option to publish the peer review history of their article (what does this mean? ). If published, this will include your full peer review and any attached files.

**Do you want your identity to be public for this peer review?** For information about this choice, including consent withdrawal, please see our Privacy Policy .

Reviewer #1: No

---

## [Editor Report · Acceptance letter]

PONE-D-24-34507R1

PLOS ONE

Dear Dr. Newport,

I'm pleased to inform you that your manuscript has been deemed suitable for publication in PLOS ONE. Congratulations! Your manuscript is now being handed over to our production team.

Kind regards,

on behalf of

Dr. Luisa Maria Diele-Viegas

Academic Editor

PLOS ONE